# Gene expression in soft-shell clam (*Mya arenaria*) transmissible cancer reveals survival mechanisms during host infection and seawater transfer

**Samuel F.M. Hart**[1,2,3], **Fiona E. S. Garrett**[1], **Jesse S. Kerr**[4], **Michael J. Metzger**[1,2]*

**1** Pacific Northwest Research Institute, Seattle, Washington, United States of America, **2** Molecular and Cellular Biology Program, University of Washington, Seattle, Washington, United States of America, **3** Department of Genome Sciences, University of Washington, Seattle, Washington, United States of America, **4** PEI Department of Fisheries, Tourism, Sport and Culture, Prince Edward Island, Canada

* metzgerm@pnri.org

## Abstract

Transmissible cancers are unique instances in which cancer cells escape their original host and spread through a population as a clonal lineage, documented in Tasmanian devils, dogs, and ten bivalve species. For a cancer to repeatedly transmit to new hosts, these lineages must evade strong barriers to transmission, notably the metastasis-like physical transfer to a new host body and rejection by that host's immune system. We quantified gene expression in a transmissible cancer lineage that has spread through the soft-shell clam (*Mya arenaria*) population to investigate potential drivers of its success as a transmissible cancer lineage, observing extensive differential expression of genes and gene pathways. We observed upregulation of genes involved with geno-toxic stress response, ribosome biogenesis and RNA processing, and downregulation of genes involved in tumor suppression, cell adhesion, and immune response. We also observe evidence that widespread genome instability affects the cancer transcriptome via gene fusions, copy number variation, and transposable element insertions. Finally, we incubated cancer cells in seawater, the presumed host-to-host transmission vector, and observed conserved responses to halt metabolism, avoid apoptosis and survive the low-nutrient environment. Interestingly, many of these responses are also present in healthy clam cells, suggesting that bivalve hemocytes may have inherent seawater survival responses that may partially explain why transmissible cancers are so common in bivalves. Overall, this study reveals multiple mechanisms this lineage may have evolved to successfully spread through the soft-shell clam population as a contagious cancer, utilizing pathways known to be conserved in human cancers as well as pathways unique to long-lived transmissible cancers.

**Data availability statement:** All code is available on GitHub (https://github.com/sfhart33/MarBTNtranscriptome), including all dependencies with version numbers. Raw sequence data are available via NCBI BioProject PRJNA874712 (https://www.ncbi.nlm.nih.gov/bioproject/874712). Data outputs can be obtained by running the supplied code on the raw data or on request. Note that code was written for our institute's working environment and thus some scripts may need to be altered manually to reproduce this analysis. Analysis was performed with an on-premises Linux server running Ubuntu 16.04. The Linux server was equipped with four Intel Xeon Gold 6148 CPUs and 250 GiB system memory. The annotated reference Mya arenaria genome can be accessed at https://www.ncbi.nlm.nih.gov/assembly/GCF_026914265.1.

**Funding:** This work was supported by the NIH (www.nih.gov). Funding includes NIH training grants T32-HG000035 and T32-GM007270 (to S.F.M.H.), career transition award K22-CA226047 and R01-CA255712 (to M.J.M). The funders did not play any role in the study design, data collection and analysis, decision to publish, or the preparation of the manuscript.

**Competing interests:** The authors have declared that no competing interests exist.

## Author summary

Transmissible cancers are rare contagious cancers that are spread by the transfer of cancer cells from one individual to another. They were reported first in dogs and Tasmanian devils, and more recently in multiple species of bivalves, though the mechanisms by which bivalve transmissible cancers have successfully spread through host populations is largely unexplored. In this study we investigate soft-shell clam transmissible cancer infection and transmission, comparing gene expression between healthy clam cells, cancer cells during infection, and cancer cells in seawater, the presumed route of transmission. We find two striking keys to the transmissible cancer's success: widespread silencing of immune processes during infection, likely facilitating escape of host immune rejection, and a conserved metabolic response to seawater, likely facilitating cell survival during transmission between hosts through the water column. Building on previous work in mammalian transmissible cancers, this highlights that silencing of immune processes is a conserved mechanism among transmissible cancers, while also discovering unique aspects of bivalve cancer adaptation and plasticity, particularly during transmission in the marine environment.

## Introduction

The maximum life span of a cancer is typically limited by the lifespan of its host, with cancer either regressing or dying along with its host. However, a small number of transmissible cancers in Tasmanian devils [1,2], dogs [3,4], and bivalves [5–10] have been able to extend their life span by transmitting to a new host like an infectious parasite. In these rare cases, cancers have gained the ability to repeatedly bypass two major barriers to cancer transmission: the physical transfer between individuals and immune rejection [11]. Transmission in devils occurs during biting and engraftment of cells on the new host's facial wounds [1], in dogs the cancer is a sexually transmitted genital tumor [3], and in bivalves the cancer cells transfer through the seawater, presumably via filter feeding [11–13]. Immunologically, the vertebrate transmissible cancers are believed to evade immune detection though mechanisms such as the downregulation of MHC genes and the release of immunosuppressive cytokines [14–17]. Additionally, it is hypothesized that low genetic diversity of the devil population and of the ancestral founder pack of dogs contributed to the ability of the cancers to initially evade immune rejection before evolving additional mechanisms [3,18]. Bivalve transmissible neoplasia (BTN) has been identified in at least ten bivalve species [5–10,19,20], indicating that bivalves may be particularly susceptible to cancer transmission. In bivalves, as in other invertebrates, there is no adaptive immune system, and it has been assumed that this contributes to the inability to uniformly reject non-self cancer cells [11]. It is unknown if there is any host innate immune response to bivalve cancers or if there are any mechanisms in the cancer that might have evolved to escape rejection by host innate immune systems.

The first species in which BTN was identified is the soft-shell clam (*Mya arenaria*), in which a single clonal lineage has spread through the native range along the east coast of North America [5]. In a previous study we analyzed *M. arenaria* BTN (MarBTN) genome sequences and found that the cancer genome was highly mutated and unstable [21]. Though this continued mutation would be expected to mediate adaptation of the cancer to its new parasitic lifestyle, it is difficult to elucidate from mutational data alone which genes and pathways are central to this parasitic ability. Here we turned to transcriptome-wide expression analysis of MarBTN to investigate the mechanisms by which it has been able to survive, proliferate, and spread through the soft-shell clam population.

## Results

### Confirmation of the hemocyte origin of MarBTN

Comprehensive annotation of all genes in the soft-shell clam genome is key to identifying expression changes in MarBTN that may have played a role in its evolution as a transmissible cancer. We previously assembled a soft-shell clam genome and annotated genes using RNAseq data from six tissues from the same clam (foot, gill, hemocytes, mantle, adductor muscle, and siphon) and genome annotation pipeline MAKER [21]. In this study, we used an improved transcriptome reference, annotated using the same genome and RNAseq data used previously (NCBI eukaryotic genome annotation pipeline). This output annotation is more comprehensive, capturing a higher number of gene models (n=44,373), transcript isoforms, exons, characterized genes, and complete BUSCOs (Benchmarking Universal Single-Copy Orthologs, a metric of transcriptome completeness) [22] (S1 Table).

We sequenced RNA from five MarBTN isolates, six tissues each from three healthy clams (hemocytes and five solid tissues), and hemocytes from an additional five healthy clams (S2 Table). We then mapped RNA reads to the new genome annotation to quantify expression for each gene. Principal component analysis (PCA) of expression across all genes separated MarBTN and hemocytes from all solid tissues across the first principal component (S1A Fig). This supports previous analyses implicating hemocytes, bivalve immune cells found in the circulatory fluid, as the likely tissue of origin for MarBTN and two independent BTNs in European cockles [21,23]. Hierarchical clustering on the top 100 tissue-specific genes also supports this origin (S1B Fig). Because BTN likely arose from a normal hemocyte, we focused on the comparison of MarBTN isolates (n=5) to healthy clam hemocytes (n=8) for the primary differential expression analysis.

### Differential expression in MarBTN compared to hemocytes

An overwhelming number of genes are significantly up- (n=8,218, 19% of annotated genes, S3 Table) or down-regulated (n=8,660, 20% of annotated genes, S4 Table) in MarBTN versus healthy hemocytes (Fig 1A), unsurprising given the clonal nature of MarBTN and their centuries of divergence from healthy clam cells. Orthologs of known human tumor suppressors (TUSC2 and RASF8, downregulated) and oncogenes (RHEB, upregulated) are among the most significant of these genes. We also see many genes involved in the cellular response to genotoxic stress, likely facilitating DNA damage repair and/or permitting cells to continue proliferating despite the ongoing genome instability observed in this lineage [21,24]. These include orthologs to PUM3 (upregulated), a gene highly expressed in some human cancers [25] that inhibits the degradation of PARP1 following genotoxic stress [26], RAD18 (upregulated), an E3 ubiquitin protein ligase involved in post-replication repair of DNA lesions [27], and ECT2 (downregulated), which is expressed during DNA synthesis and can lead to genotoxic stress-induced cell death [28]. Among genes with the largest fold difference in MarBTN versus healthy hemocytes we see genes involved in cell adhesion (TENX upregulated, CNTN5 downregulated). This likely contributes to MarBTN's distinctive non-adhesive spherical phenotype and may facilitate the hyper-metastatic ability of MarBTN to engraft and release from tissues repeatedly. An innate immune signaling gene, GBP1, is also among the most downregulated and could play a role in the cancer's ability to evade host immune rejection of non-self cells, an open question across all transmissible cancers. Thousands of other genes are highly mis-regulated and likely to play important roles in MarBTN, but most of these are either uncharacterized or do not have an obvious link to cancer. This is not unexpected, since in addition to known cancer-associated genes, we would expect this set to include undiscovered cancer-associated genes, genes specific to bivalve oncogenesis, genes specific to transmissible

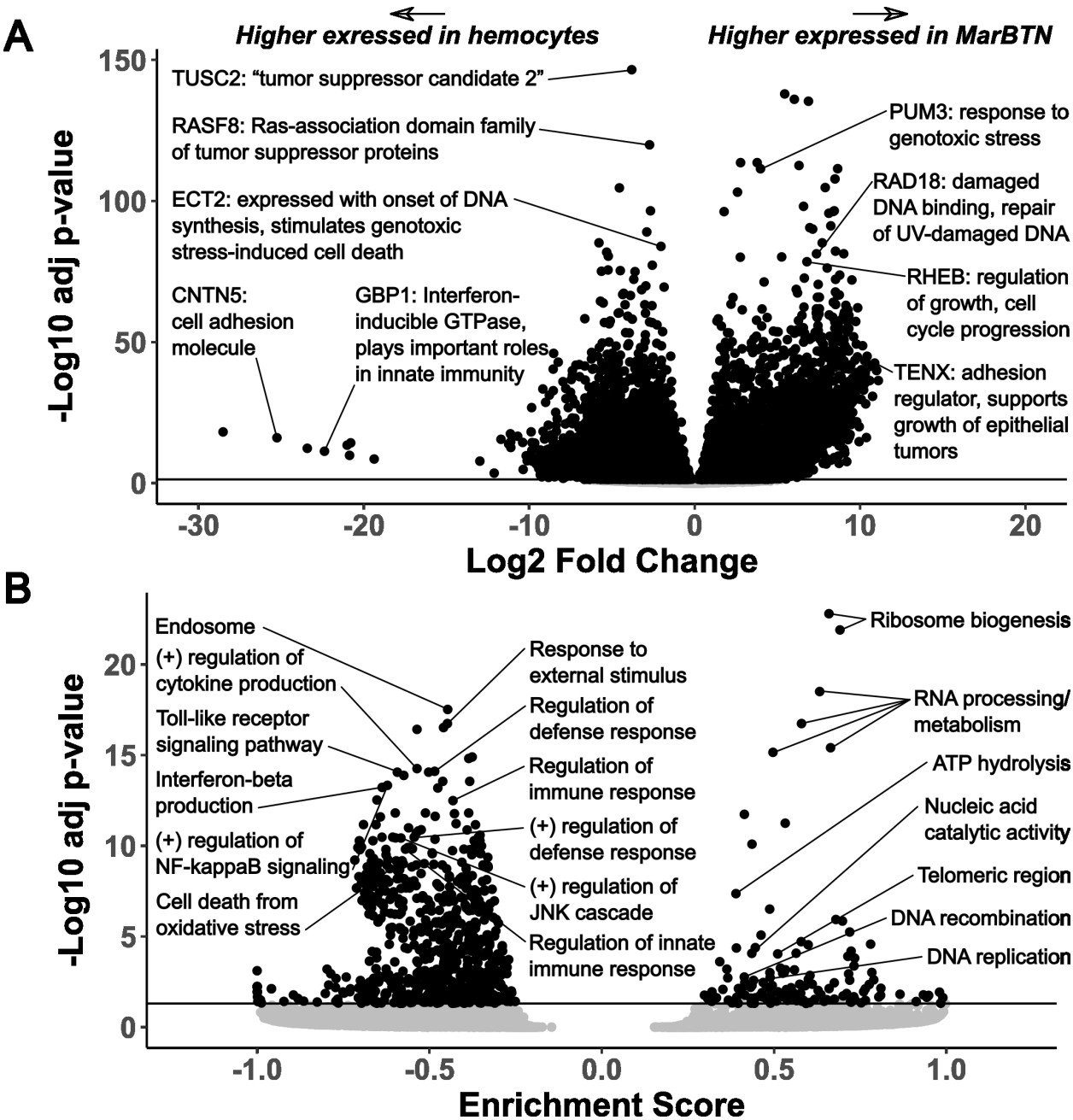

**Fig 1. Top differentially expressed genes and pathways in MarBTN.** Volcano plots of differentially expressed genes (**A**) and gene sets (**B**) from the comparison of MarBTN isolates (n=5) versus healthy hemocytes (n=8). Genes of note are labeled with annotations and abbreviated descriptions. Line marks false discovery rate adjusted significance threshold (p < 0.05), with genes below threshold colored in grey. Note grey points are largely not visible in A due to axis scale. "(+)" = positive regulation.

cancer cell survival, and genes that do not provide a selective advantage but are differentially regulated either by chance or as a byproduct of selection on genes in related pathways.

To investigate transcriptome-wide expression trends we turned to gene set enrichment analysis (GSEA), which ranks all genes by significance of differential expression and then tests whether subsets of those genes involved in a particular process, function or localization

are disproportionately up- or down-regulated [29] (e.g., S2 Fig). Of 9,453 pathways tested, we observed 135 significantly upregulated pathways and 756 significantly downregulated pathways (Fig 1B). The most highly upregulated pathways involved RNA processing and ribosome biogenesis (S5 Table), which are recognized as important for cell growth and proliferation of cancer cells [30]. ATP hydrolysis and DNA replication/recombination are also among the top pathways and would be key for a metabolically demanding growth and division of cancer cells. We also observe upregulation of genes whose products localize to telomeric regions and DNA repair complexes, perhaps facilitating maintenance of genome integrity in response to damage and telomere shortening, which would be critical for the survival of a long-lived transmissible cancer.

Interestingly, the top downregulated pathways all relate to immune responses, such as cytokine production, NF-κB activation, toll-like receptor signaling and defense/inflammatory/ innate immune responses (S6 Table). We suspected this may be an evolved response allowing MarBTN to better evade host immune rejection, though alternatively it could represent the downregulation of pathways that are needed in the immune cell type that the cancer arose from but are now unnecessary in a cancer, or the cancer arising from an earlier differentiation stage of hemocyte than the mature hemocytes analyzed here. To test whether this finding was due to downregulation of hemocyte-specific genes or due to downregulation of immune genes expressed in most cell types, we looked at differential expression comparing MarBTN isolates (n=5) to solid tissues (n=15: 5 tissues each from 3 clams). Many of the same genes and pathways were similarly up- or down-regulated as they were in the hemocyte comparison (S3 Fig), with immune pathways continuing to dominate the downregulated gene pathways. This indicates the observed immune downregulation is not primarily due to the comparison to hemocytes and instead supports the hypothesis that this is a mechanism to evade host immune rejection. To test whether this observation of widespread immune downregulation could be validated using an alternative method, we looked at genes known to cause inborn errors in immunity when mutated [31], finding that immunodeficiency genes found in clams (n=293) were much more likely to be downregulated than expected by chance (p=8e-9, two-tailed one-sample t-test, S4A Fig). In addition to immune pathways, we observe downregulation of stress responses such as the JNK/MAPK cascades and oxidative stress-induced cell death. These pathways likely contribute to the ability of MarBTN to survive repeated exposure to the extreme environments of hypoxic late-stage cancer infections [32] while continuing to proliferate and maintain the ability to infect new hosts.

Although we observe many differentially regulated genes and pathways in MarBTN, these samples still represent a single transmissible cancer lineage and therefore an effective sample size of one. To investigate conserved trends across BTNs, we compared our results to those from a recent study by Burioli et al. [33] in an independent BTN lineage that originated in *Mytilus trossulus* and circulates in various *Mytilus* species (MtrBTN2). Burioli et al. similarly compared gene expression of BTN to healthy hemocytes, finding proliferation, metabolic, cell fate, DNA repair, adhesion, and immune pathways were differentially regulated. To identify convergent evolution between MarBTN and MtrBTN2, we identified genes that were significantly differentially regulated in both cancers and shared an exact gene annotation match (S7 Table, n = 1498). More genes were either upregulated in both cancers (n = 373) or downregulated in both cancers (n = 569) than would be expected by chance (942/1498, 63%, p = 2e-23, Chi-squared test, df = 1). This indicates that some of the same genes may be playing a role in both cancers, particularly genes that are downregulated in both cancers, where the greatest overlap was observed. The genes with the strongest downregulation in both the transmissible cancers from clams and mussels included genes involved in the innate immune inflammatory response (toll-like receptors, ficolin-2), cell cycle regulation (cell division control protein 42),

stress response (heat shock protein beta-1) and apoptosis (caspase-3). As more data become available from other BTN lineages, further analysis more thoroughly identifying gene homology across bivalves and comparing differentially expressed genes in their respective BTNs may help us to zero on in universally conserved mechanisms that have repeatedly evolved to allow BTNs to survive, repeatedly engraft in new animals, and evade the host response.

### Genome instability affects gene expression

We previously observed that MarBTN's genome is highly unstable, displaying widespread genome rearrangement, copy number gains, and transposable element activity [21]. With gene expression data, we were interested to investigate how this genome instability affected the cancer's transcriptome, as the intermediary between genotype and phenotype. We first quantified the number of fusion transcripts in each sample, as structural mutations would be expected to generate gene fusions that may play important roles in MarBTN evolution. We observed ~10-fold more gene fusions in MarBTN isolates than the baseline number observed in healthy hemocyte samples (Fig 2A, avg = 416 vs 39, p = 1.5e-4, two tailed t-test with unequal variance). In addition to true fusions generated by germline polymorphisms, the small number of fusions in healthy samples may be due to genome mapping errors, transcript read-throughs, transposable elements missed in masking, or structural variants polymorphic in the clam population, while the increased number of fusions in cancer samples is likely caused by somatic genome rearrangement. Fusions found in all cancer samples but no healthy samples (n=181, S8 Table) include fusions from early in the cancer's somatic evolution that may have contributed to the oncogenesis and/or transmission ability of the lineage.

Copy number alteration is a known mechanism in cancers to alter the expression of cancer-promoting genes [34]. To test whether copy number affects expression in MarBTN we binned genes by genomic copy number, observing that MarBTN expression relative to healthy

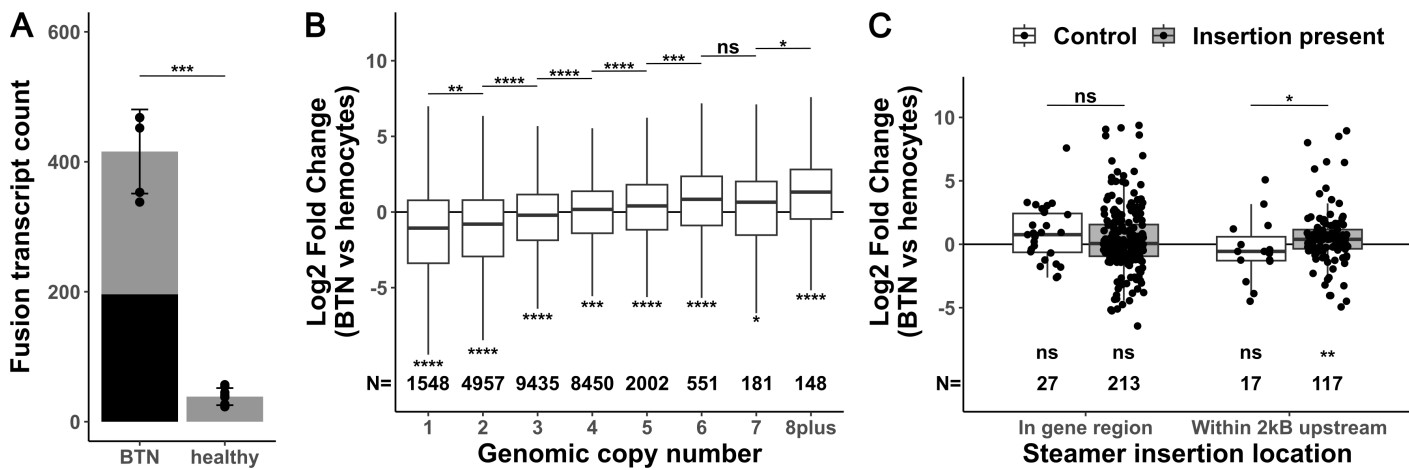

**Fig 2. Fusions, copy number and transposable element insertions influence gene expression.** (**A**) Number of fusion transcripts per sample (dots), with mean and standard deviation for MarBTN (n=5) and healthy hemocyte (n=8) sample groupings. Black bar represents fusions found in all MarBTN samples (196). Statistical test is two-tailed t-test with unequal variance. (**B**) Log fold change in expression of MarBTN versus healthy hemocytes, binning genes by genomic copy number for each gene. Upper statistical tests are two-sided Wilcoxon rank-sum tests comparing medians between adjacent copy numbers, lower statistical tests are one-sample two-sided Wilcoxon rank-sum tests comparing medians versus no change. Gene counts for each group are listed below box plots. (**C**) Log fold change in expression of MarBTN versus healthy hemocytes for genes with a *Steamer* insertion within the gene itself or in the 2kB region upstream of the gene. Each set shows genes with *Steamer* insertions present in these samples (grey box) and, as a control, genes with no *Steamer* insertion present in these samples but at which *Steamer* insertion sites have been observed in a different sub-lineage of MarBTN (white box). Statistical tests and counts are the same as listed in (B). ns: not significant, *: p<0.05, **: p<0.005, ***: p<0.0005, ****: p<0.00005.

hemocytes scales with copy number state (Fig 2B). MarBTN from New England, USA has an average ploidy of ~3.5N across the genome, with >80% of genome at 2-4N [21], and we see that median relative expression of genes ≥4N is higher than average while lower than average for genes ≤3N (p<0.05 for all copy number states, one sample two-sided Wilcoxon rank-sum test). Given the widespread copy number changes in the MarBTN genome and ongoing instability, gain or loss of gene copies likely represents a mechanism that has helped scale expression of key genes for MarBTN to adapt as a transmissible cancer.

We were also interested in whether transposable element activity influences the expression of nearby genes, so we looked at the expression of genes near insertions of the LTR-retrotransposon *Steamer* (Fig 2C), one of the most active and best characterized transposable elements in MarBTN [35]. When analyzing genes in which *Steamer* has inserted into the gene region, we see no significant increase in expression, compared to healthy hemocytes, which would not have the *Steamer* insertion (p = 0.15, one-sample two-sided Wilcoxon rank-sum test). However, we previously found that *Steamer* preferentially inserts upstream of genes [21], and here we find that expression in genes with insertions 0-2kB upstream were higher than those genes in healthy hemocytes without the insertion (p = 0.0034, one-sample two-sided Wilcoxon rank-sum test), likely due to promotors or enhancers in *Steamer*'s LTR [36]. To control for the possibility this bias was due to an insertion preference for highly expressed genes due to accessible chromatin (instead of the insertion causing the expression change itself), we also looked at the expression of genes lacking *Steamer* insertion sites in MarBTN samples analyzed here, but where *Steamer* insertions had been observed in a different sub-lineage of MarBTN (found in clams in Prince Edward Island, Canada). When compared to differential expression of this control set of known *Steamer*-accessible genes instead of just to healthy hemocytes, we observe the same pattern: insertions upstream of genes also had a significant effect on expression (p= 0.038), and insertions in gene regions still did not have an effect (p = 0.27, two-sided Wilcoxon rank-sum test).

Overall, we see that gene fusions, copy number alterations, and *Steamer* insertions all influence expression and thus likely contributed to the adaptability of this lineage as a transmissible cancer. Indeed, we see that some of the top individual genes differentially regulated in MarBTN have these mutations (S4B-E Fig). These include an upregulated high copy number methyltransferase (METL2, CN=6, LFC=3.4, p=6e-97), loss of five of the eight most downregulated genes (adhesion gene CNTN5, immune gene GBP1, and three uncharacterized genes, CN=0, LFC <-1.5e6), a *Steamer* insertion upstream of the upregulated carbohydrate sulfotransferase CHSTF (LFC=257, p=6e-77), and upregulation of an uncharacterized gene (LFC=346, p=3e-97) fused upstream of cell cycle control gene CC14A. However, most top differentially regulated genes are not directly affected by one of these structural mutation types, indicating that most differential expression is driven by other mutation types, epigenetic modifications, or indirect effects of mutated regulatory genes.

## Transcriptomic plasticity in response to saltwater

The late stage of MarBTN infection is one in which a highly pure sample can be obtained for sequencing, but the MarBTN infection cycle also includes transmission to engraft and proliferate in a new clam host through repeated metastasis-like jumps (Fig 3A). This transfer is believed to occur through release of cells into seawater and uptake by filter-feeding, an inference supported by findings that MarBTN cells survive for weeks in saltwater and that MarBTN-specific DNA can be detected in tank water where MarBTN-infected clams are maintained [12]. This metastatic transmission stage would involve a different environment and selective pressures than those faced during infection, and it is possible that MarBTN has evolved the plasticity to respond to the two stages differently.

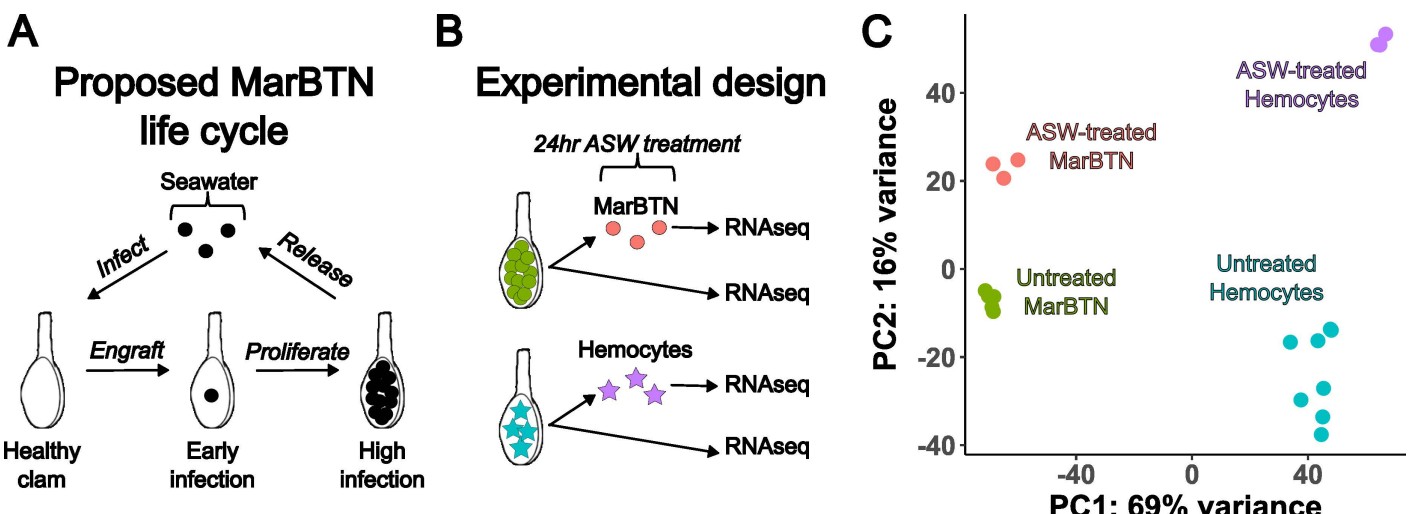

**Fig 3. Transcriptomic response to seawater exposure.** (**A**) Proposed life cycle for MarBTN infections and (**B**) experimental design to investigate gene expression during seawater transmission. (**C**) Principal component analysis results from gene expression across all genes for ASW-treated and untreated MarBTN and hemocytes. ASW = artificial sea water.

To test this possibility, we incubated an aliquot of three MarBTN isolates in artificial sea water (ASW) for 24 hours prior to RNA sequencing to investigate the gene expression response in this stage compared to direct RNA sequencing of another aliquot of the same isolate (Fig 3B). As a control to test for the intrinsic response of clam cells to seawater, we also exposed three healthy hemocyte isolates to ASW before sequencing. We performed principal component analysis on the gene expression results of these samples, with the primary principal component separating hemocytes from MarBTN and the secondary principal component separating ASW-treated cells from untreated cells (Fig 3C). Gene expression was more similar within treatment groups (ASW-treated versus pre-treatment) than source clam pairings (biological replicates before and after treatment) indicating that saltwater exposure results in a consistent transcriptomic response greater than the biological variation among our samples (S5 Fig). We compared ASW-treated vs. untreated MarBTN and ASW-treated vs. untreated hemocytes for differentially expressed genes and gene sets, dividing results into two groups: differentially regulated in both comparisons (Fig 4A and 4B) and differentially regulated in MarBTN but not hemocytes (Fig 4C and 4D). Genes differentially regulated in both comparisons would indicate intrinsic responses to seawater conserved by MarBTN and healthy clam hemocytes, while genes differentially regulated in MarBTN but not hemocytes would indicate MarBTN-specific responses to seawater that may be adaptive in the cancer.

Among conserved gene responses (5,218 of 44,373 genes), the outlier upregulated gene was *PCKGC*, the main control point for the regulation of gluconeogenesis, likely representing a metabolic response to the new energy-source-free environment. Similar gluconeogenesis-activating responses have been observed in glucose-deprived human cancer cells [37]. The second most significant upregulated gene is SPT13, a gene involved in cell migration. Interestingly, this gene is fused upstream to SLC5A9, a sodium/glucose cotransporter, in MarBTN, perhaps leading to SPT13 to be more highly upregulated in MarBTN (LFC=34) than hemocytes (LFC=5.4) in response to seawater (S4 Fig). Another notable gene from the top upregulated genes (S9 Table) is *XIAP*, which is part of a family of apoptotic suppressor proteins and likely helps cells to avoid an apoptotic response to seawater. *XIAP* also modulates inflammatory and immune signaling via NF-kappaB and JNK activation [38], indicating these

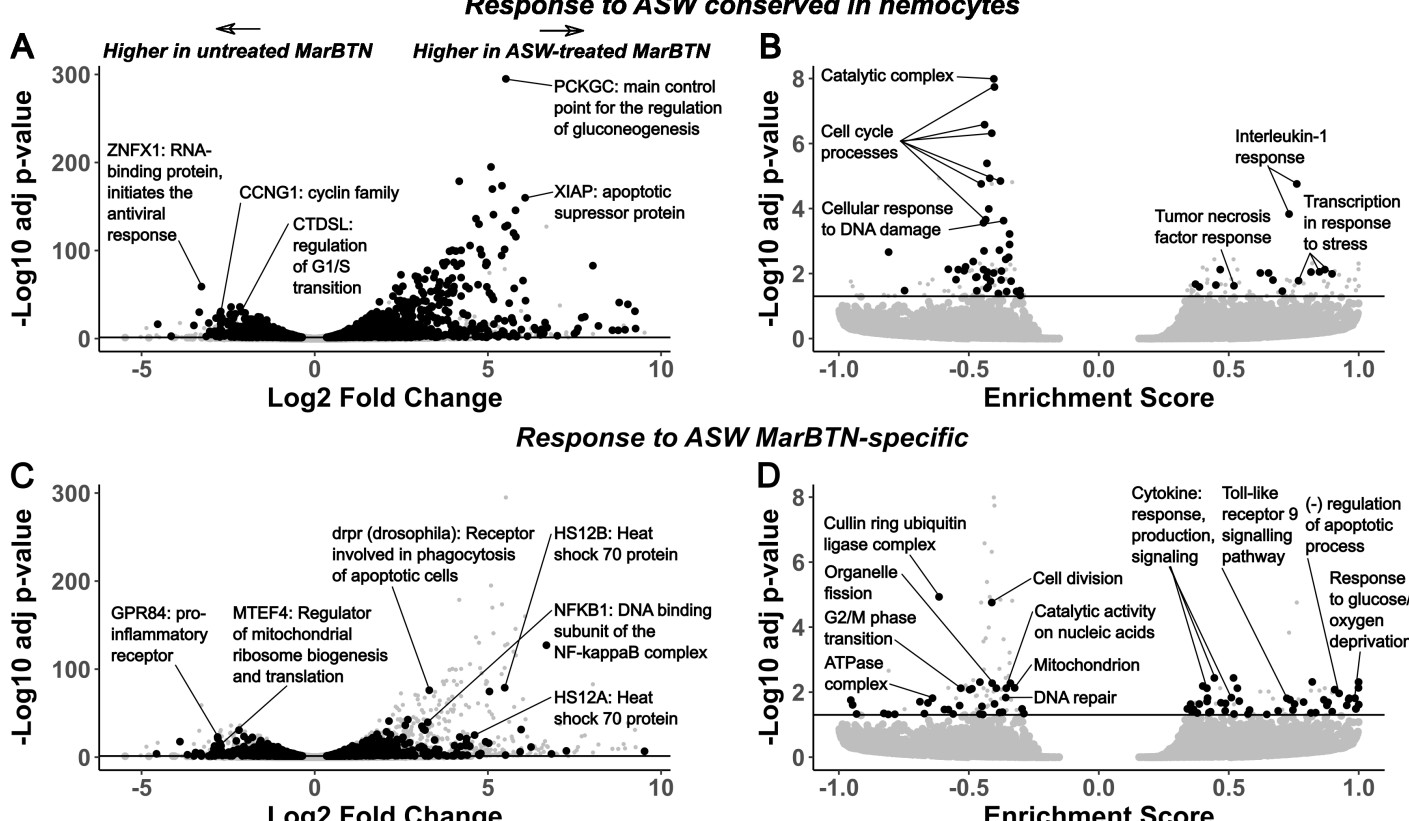

**Fig 4. Top differentially expressed genes and pathways in MarBTN after exposure to seawater.** Volcano plots of fold change and significance of differentially expressed genes (**A/C**) and gene sets (**B/D**) in ASW-treated MarBTN (n=3) versus untreated MarBTN (n=3). Genes of note are labeled with annotations and abbreviated descriptions. Volcano plots are filtered for genes/sets that are (**A/B**) or are not (**C/D**) differentially expressed in ASW-treated hemocytes (n=3) versus untreated hemocytes (n=3). For comparison, the reciprocal comparison is included in small grey points on each plot. Line marks false discovery rate adjusted significance threshold (p < 0.05), with genes below threshold colored in grey.

pathways, which were downregulated when comparing untreated MarBTN to healthy hemocytes, may be activated in both MarBTN and hemocytes in response to seawater. Indeed, when we use GSEA to identify differentially regulated pathways (n=150 of 9,423 gene sets), we see pathways involved in the response to interleukin-1, tumor necrosis factor, and stress among the conserved upregulated pathways (S10 Table), indicating these pathways are likely to be intrinsic responses of clam cells when exposed to seawater.

The top conserved downregulated gene is *ZNFX*, which encodes an RNA-binding protein involved in antiviral response [39], though it is unclear why this gene might be lower expressed in seawater. Among the other top conserved downregulated genes (S11 Table) are *CCNG1*, a member of the cell cycle controlling cyclin family [40], and *CTDSL*, which is involved regulating the G1/S transition [41]. Many of the top downregulated pathways are also involved with cell cycle progression (S12 Table), likely representing mechanisms to halt proliferation in the absence of host nutrients and may help all cells to survive the seawater environment by entering a quiescent state. These conserved responses to seawater could represent a starting point that could be built upon during MarBTN evolution and selection for transmission ability.

Although the most significant differentially regulated gene and gene sets were conserved, many genes (n=1,995 of 44,373 genes, Fig 4C) and gene sets (n=89 of 9423 gene sets, Fig 4D)

were differentially regulated in response to seawater in MarBTN but not hemocytes, indicating MarBTN may have evolved additional mechanisms to survive seawater transfer. Among the top upregulated genes (S13 Table) are two heat shock protein family A paralogs (HS12A, HS12B), which can help protect cells from heat, cold, hypoxia or low glucose [42] and may be helping MarBTN cells survive one of these seawater extremes. Glucose/oxygen deprivation response and negative regulation of apoptosis are among the top upregulated gene sets (S14 Table), indicating that MarBTN has a broad response, not found in healthy hemocytes, to survive these aspects of seawater exposure.

Interestingly, several gene sets that are upregulated after ASW exposure specifically in MarBTN are immune response pathways (n=13 of 56), including cytokine production/response/signaling and toll-like receptor 9 signaling. This is an interesting reversal of the downregulation observed in the same pathways comparing untreated MarBTN cells to healthy hemocytes, although these pathways are still lower expressed in ASW-treated MarBTN than healthy hemocytes (S6 Fig). This could indicate that the downregulation of these immune pathways may not be as important outside the context of a host immune system, that some innate immune system processes are reactivated in response to pathogens outside of host, or that these processes are required for some aspect of MarBTN-specific survival/engraftment.

The outlier MarBTN-specific downregulated gene sets (S15 and S16 Tables) are cell division and the cullin-RING ubiquitin ligase complex, which controls cell cycle progression and other cellular processes [43]. Other metabolic pathways are downregulated in MarBTN but not hemocytes, such as catalytic activity on nucleic acids, organelle fission, ATPase complex localization, and mitochondrial localization. These responses may reflect metabolic processes that are active in proliferative MarBTN, but not hemocytes, that are shut down in response to seawater exposure. However, of the metabolic pathways listed, only catalytic activity on nucleic acids was upregulated in the initial MarBTN versus hemocytes comparison (S5 Fig). Alternatively, the other metabolic pathways may represent additional mechanisms that MarBTN has evolved to halt metabolism in response to nutrient-poor seawater that do not exist in hemocytes. This experiment reveals consistent MarBTN-specific seawater responses that likely facilitate transfer to new hosts, aided by the cells' inherent plasticity to respond to seawater inherited from its hemocyte origins.

## Discussion

All cancers must evolve to evade intrinsic and extrinsic barriers to successfully develop as a cancer [44]. In addition to overcoming these barriers, transmissible cancers also evolve to repeatedly transfer to new hosts and proliferate despite anti-tumor and non-self rejection mechanisms [11]. This all occurs while having no evolutionary history as a transmitting parasite prior to oncogenesis [45]. By analyzing the MarBTN transcriptome during infection and transfer, we identify possible mechanisms by which this transmissible cancer has adapted to overcome these barriers, most notably the widespread downregulation of immune signaling pathways when in hosts and survival responses to seawater exposure.

We observe mis-regulation of many gene types in MarBTN that would be expected in any cancer, such as genes involved in metabolism, cell cycle progression, adhesion, tumor suppression, genome instability and immune evasion [46]. The downregulated biological processes overwhelmingly relate to immune signaling functions (Fig 1B) and likely represent an adaptive mechanism to repeatedly evade host detection/rejection as MarBTN spread through the soft-shell clam population. Innate immune-related biological processes were also significantly downregulated in a mussel transmissible cancer [33], while the mammalian transmissible cancers display MHC downregulation [14–16]. Together this indicates that downregulation of immune processes is a conserved mechanism among transmissible cancers,

though which processes likely depend on the host context and whether an adaptive immune system is present. As more BTNs are identified and characterized, a systematic comparison of differentially expressed genes and pathways would likely identify additional examples of convergent evolution and reveal underlying mechanisms of transmissible cancer evolution. Such mechanisms may also highlight more generally how conventional cancers evolve to evade innate immune responses or promote metastasis, since all BTNs have strong selective pressure for repeated metastasis.

Overcoming barriers to repeated transmission events and challenge by new host immune systems would suggest a highly adaptable cellular lineage. Indeed, widespread mutation and genome instability were observed in our prior MarBTN genomics study [21], and here we observe cases in which that genome instability directly affects the cancer transcriptome. Copy number alterations, which are highly variable across BTNs [21,23,47], may represent a particularly malleable mutation type for fine-tuning gene expression up or down to maximize cancer fitness in the face of changing selective pressures. Examples of the expressed genes influencing genome instability are also apparent, such as upregulation of genotoxic stress response gene *PUM3*. Previous work also identified the upregulation of an error-prone polymerase (*POLN*) and upregulation of *HSP9* (mortalin), which has been shown to sequester DNA damage response molecule p53 [21,48]. This tolerance of genome instability, in combination with the generation of innovative mutations that affect gene expression, creates prime conditions for MarBTN to adapt and spread as a transmissible cancer. This cancer has successfully spread for at least 200 years [21], but it remains to be seen whether this lineage can continue to survive with widespread genome instability and mutation, or whether adaptability is solely a short-term benefit with the long-term cost of deleterious mutation accumulation in an asexual lineage, the process known as Mueller's ratchet [49].

In our seawater exposure experiment, we were surprised that the strongest responses, involved in metabolism, stress response, and cell cycle arrest, were conserved in both cancer and healthy clam hemocytes. A recent paper observed that mussel (*Mytilus edulis desolationis*) hemocytes are regularly released into seawater and transfer live into new mussels, postulating that this may facilitate the transfer of pathogen infection via hemocytes themselves [50]. This raises the intriguing possibility that bivalve hemocytes, for some unknown reason, are already adapted to survive for extended periods of time outside the bivalve body. Since BTNs originated as hemocytes [21,23], this would mean they already have the inherent ability to survive in seawater and enter new hosts and may in part explain why transmissible cancers are so common in bivalves. On top of these conserved responses, we observe cancer-specific responses to seawater that are absent in hemocytes, indicating that MarBTN may have evolved additional genetic or epigenetic plasticity mechanisms to increase its transmission ability.

In this study we investigated gene expression at two key stages of the hypothesized MarBTN life cycle: late-stage cancer infection and saltwater transfer. To gain a comprehensive understanding of MarBTN infection and progression, future work should also investigate gene expression at the early stages of cancer engraftment and proliferation, which would require sorting MarBTN from host cells. Host cell gene expression would also be informative about the clam defense response to MarBTN infection, and what defense regimens succeed at keeping the cancer contained versus succumbing to the infection. BTNs appear to be a common occurrence in bivalve populations and are likely to impose a strong selective pressure for resistance [45,51]. Identification of innate immune system cancer resistance mechanisms of hosts and countering evasion mechanisms in transmissible cancers, selected for by repeated infection, may each have broader implications in our understanding of the host-pathogen relationship of conventional cancers.

## Methods

### Genome annotation

To utilize the NCBI Eukaryotic genome pipeline we supplied NCBI with the previously assembled *M. arenaria* genome [21] and RNAseq data for six tissues (foot, gill, hemocytes, mantle, adductor muscle, and siphon) from the clam that was used to assemble the reference genome. The output genome and annotation can be found at https://www.ncbi.nlm.nih.gov/assembly/GCF_026914265.1. We compared the completeness of the NCBI genome annotation to the original MAKER-annotated genome with Benchmark of Universal Single Copy Orthologs (BUSCO v3 [22]) using the command: busco -m prot -l metazoa_odb10 and calculated other stats in S1 Table using custom scripts.

### Sample collection

Clams were collected in Prince Edward Island, Canada (PSH samples, healthy clams) or by a commercial shellfish supplier in Maine (MELC and FFM samples, healthy and cancerous clams) and shipped live on ice to the Pacific Northwest Research Institute in Seattle, WA (S2 Table). Upon arrival, hemolymph was drawn from the pericardial sinus and checked for the presence of MarBTN with a highly sensitive cancer-specific qPCR assay (as described in [12]). The selected healthy clams were undetectable for the cancer-specific qPCR marker, while the selected MarBTN-infected clams had only cancerous cells (no host hemocytes) visible in hemolymph under a microscope. From healthy clams, 1 mL of hemolymph was spun at $500 \times$ g for 10 min at 4 °C and hemolymph was pipetted off to leave a hemocyte cell pellet. For three of the healthy clams, dissections were performed to isolate foot, gill, mantle, adductor muscle, and siphon tissues. In MarBTN-infected clams MarBTN cells have been shown to be non-adherent, while hemocytes will adhere to a plate within an hour [12,24]. Therefore, to further purify the MarBTN cells, 1 mL of hemolymph from MarBTN-infected clams was left for 1 hour in a 24-well plate at 4 °C to allow host hemocytes to adhere to the plate and the non-adherent MarBTN cells were collected by pipette. These isolates were spun at $500 \times$ g for 10 min at 4 °C and hemolymph was removed to leave a MarBTN cell pellet. For three MarBTN isolates, half of the cells were resuspended in artificial sea water (ASW, 36 g/L Instant Ocean, Blacksburg, VA, USA) with antibiotics (1× concentration of penicillin/streptomycin, GenClone: Genesee Scientific, and 1 mM voriconazole, Acros Organics: Thermo Fisher Scientific) as described in [12], incubated at 4 °C for 24 hours to simulate seawater transfer, spun at $500 \times$ g for 10 min at 4 °C, and hemolymph was pipetted off to leave ASW-treated MarBTN cell pellets. For three healthy hemocyte isolates, which are adherent, half of the cells (0.5mL) for each sample were left to adhere to a 24-well plate for 1 hour before pipetting off hemolymph, adding ASW plus antibiotics as above, incubating at 4 °C for 24 hours, pipetting off ASW, proceeding directly with RNA extraction with the digestion step directly on the plate that cells were adhered to. All other samples (healthy hemocytes, tissues, MarBTN isolates, and ASW-treated MarBTN isolates) were covered in RNAlater and stored at −80 °C until RNA extraction.

### RNA extraction

RNA was extracted from each sample using the Qiagen RNeasy kit (Qiagen, Hilden, Germany), eluting in 60 μL elution buffer. Solid tissues were homogenized with a disposable plastic mortar and pestle in liquid nitrogen prior to extraction. DNase I (2 μL, 2,000 U/ml, RNase-free, New England Biolabs, Ipswich, MA), 10× DNase buffer, and water were then added to the eluted RNA to a total of 100 μL, and the reaction was incubated for 1 h at room temperature. Then 250 μL ethanol was added and mixed by pipette, and it was added to a second Qiagen RNeasy column. The RNeasy protocol was followed, skipping the RW1 step,

adding 500 μL RPE 2×, and eluting in 40 μL elution buffer. RNA samples were then sequenced on a single Illumina HiSeq 4000 lane for 20–30 million 150 bp paired-end reads per sample (Genewiz, Leipzig, Germany).

## Differential expression analysis

We indexed the annotated genome and aligned reads for all samples using STAR [52], quantifying reads mapped per gene using --quantMode GeneCounts. We confirmed MarBTN isolates were all part of the USA sub-lineage at 48/48 mitochondrial loci differentiating USA vs PEI (see [21]), and the VAFs of USA-specific mitochondrial SNVs were 96–99+% in all samples, confirming high BTN purity.

We merged counts per gene for all samples and ran DESeq2 [53], using sample groupings (healthy tissues, hemocytes, or MarBTN) as conditions on which to test differential expression for all gene models (n=44,373). Note that DESeq includes a library size normalization, so all significance values are adjusted for this normalization. We performed principal component analysis by applying variance stabilizing transformation using vst() and then plotPCA() from the DESeq2 package. We determined the top tissue-specific genes for each tissue by comparing each to the five others (e.g., gills versus all five non-gill tissues) using DESeq2 on read counts per gene, sorting by the "stat" output and taking the top 100 overexpressed genes for each tissue. We normalized read counts for each sample by calculating total mapped reads and multiplying so that each sample totaled the same number of reads as the maximum sample (note this is upstream DEseq on read counts, not sample group comparisons). We then performed hierarchical clustering on expression of the combined 6 sets of 100 top overexpressed genes for each tissue using the pheatmap package with clustering_distance_cols = "canberra". ASW-treated samples were excluded from the original clustering analysis (S1 Fig), then included alongside untreated hemocytes and MarBTN for principal component analysis and hierarchical clustering using expression of all genes and the same packages/functions as described above (S5 Fig).

For the comparison of MarBTN to solid tissues, we combined all five solid tissue types and ran DESeq2 versus MarBTN. We ran similar comparisons for ASW-treated versus untreated MarBTN and hemocytes. For the comparison of differential expression results from multiple DESeq2 runs (e.g., S3 Fig) we calculated a "+/- directional" p-value by taking the -log10 of the adjusted p-value when the log2 fold change was positive and log10 of the adjusted p-value when the log2 fold change was negative.

## Gene set enrichment analysis

For gene set enrichment analysis, we first had to determine gene sets for *M. arenaria* genes. We used blastp to determine the closest uniprot hit for each gene, taking the gene with the highest e-value and keeping genes that have a hit>1e-6 (24,145 or 56% of genes). We then merged this list of genes with the msigdbr [54] *Homo sapiens* ontology gene set ("C5") to get putative *M. arenaria* gene sets (S17 Table). Separately, genes were rank-ordered using "stat" DESeq2 parameter using ties.method = "random" for each comparison (MarBTN vs hemocytes, MarBTN vs solid tissues, ASW-treated MarBTN vs untreated MarBTN, etc.). The "stat" parameter ranking corresponds directly to significance ranking, but with positive and negative values for up/down regulated genes. We then ran GSEA (clusterProfiler package) on each ranked gene lists with additional parameters: eps = 1e-1000, pvalueCutoff = 1, seed = 12345.

## Comparison to other data sets

To compare differentially expressed genes with different data sets, we merged our genes with genes from Burioli et al. [33] and a list of immunodeficiency genes as defined by the Inborn

Errors of Immunity Committee (https://iuis.org/committees/iei/, latest update from October 2024). We only kept instances of an exact gene annotation match to our closest uniprot hit, as identified in the GSEA section.

### Identification of fusion genes

We identified fusion transcripts using STAR-Fusion (v1.11.0 [55,56]). We first generated a custom genome index using prep_genome_lib.pl on the annotated genome with "--pfam_db current --dfam_db human" as default run parameters. We then ran STAR-Fusion each sample individually with default setting plus additional parameters: --FusionInspector validate, --examine_coding_effect, --denovo_reconstruct. We determined fusions shared by multiple samples by identical left and right breakpoints, excluding fusions that were found in all samples (n=16) as likely genome assembly or annotation artifacts from our results. To compare the number of fusions in MarBTN samples versus hemocytes, we used a two-tailed t-test with unequal variance.

### Copy number effects

Genomic copy number calls were determined in 100 kB segments for USA sub-lineage MarBTN samples in previous work [21]. The copy number regions were observed to be nearly identical between the samples of the MarBTN from the USA sub-lineage, so while there are likely minor differences in these samples, these copy number calls are likely to be similar for the samples of this current study. We used bedtools intersect to link each gene to its genomic copy number state, excluding genes that were not at >90% at a single copy number state (e.g., gene spans a breakpoint in copy number). We dropped CN0 due to issues with reliably calling CN0 vs off-target mapping due to repetitive regions. We then created a boxplot for each copy number state of the log2 fold change of MarBTN versus healthy hemocytes, observing that higher copy number genes tend to have increased expression versus their diploid healthy references. We applied two tests for significance: two-sided Wilcoxon rank-sum tests between adjacent copy numbers and one-sample two-sided Wilcoxon rank-sum tests versus no fold change (log2 fold change = 0).

### *Steamer* insertion effects

We had also determined *Steamer* insertion sites in previous work [21], and assumed that insertions previously found in all USA sub-lineage MarBTN samples would also be present in the samples of this current study, which were all confirmed to be from the USA sub-lineage (see above). We determined where *Steamer* had inserted within genes or within 2 kB upstream genes using bedtools intersect. As a control, we took genes intersecting insertions that were found in the PEI sub-lineage but not USA sub-lineage as sites that are unlikely to be present in the samples of this current study but that were accessible for *Steamer* insertion. We applied two tests for significance: two-sided Wilcoxon rank-sum tests between genes with *Steamer* insertions and controls, and one-sample two-sided Wilcoxon rank-sum tests versus no fold change (log2 fold change = 0) for genes with *Steamer* insertions.

### Supporting information

**S1 Fig. Hemocyte origin of MarBTN supported by PCA and clustering with new gene annotations.** (**A**) Principal component analysis of normalized expression across all genes, with PC1 separating MarBTN and hemocytes from all other tissues. (**B**) Hierarchical clustering of all RNA sequenced samples (excluding ASW-treated samples) by the expression of the top 100 most significant genes expressed in each specific healthy tissue relative to all other

tissues, with heatmap of normalized relative gene expression for each gene. MarBTN ("BTN") clusters most closely with hemocytes ("heme"), supporting principal component analysis results. Results for both panels closely match similar analyses with previous genome annotation and a smaller sample set [21].
(TIF)

**S2 Fig. Example GSEA results for one each of the top up- and downregulated pathways.** Running enrichment score (green), which increases each time it hits a gene in the gene set (black bars along x-axis) for the ribosome biogenesis biological process (**A**), one of the top upregulated pathways, and positive regulation of innate immune response biological process (**B**), one of the top downregulated pathways. Red dotted line marks the peak enrichment score, corresponding to the x-axis of volcano plots. Genes were rank-ordered using "stat" DESeq2 parameter, which corresponds directly to significance ranking, but with positive and negative values for up/down regulated genes.
(TIF)

**S3 Fig. Differential expression is similar whether comparing MarBTN to hemocytes or solid tissue.** Adjusted p-values, further adjusted to be positive for upregulation and negative for downregulation, from MarBTN versus healthy hemocytes differential expression results (x-axes) and MarBTN versus solid tissues differential expression results (y-axes) for individual genes (**A**) and gene pathways (**B**). In general, genes and pathways that are upregulated versus hemocytes are also upregulated versus solid tissues, indicating that major differential expression results and conclusions are not artifacts of the comparison with hemocytes. Lines represent false discovery-corrected $p < 0.05$ significance thresholds.
(TIF)

**S4 Fig. Volcano plots showing differential expression analysis comparing BTN vs hemocytes (A-E, as seen in Fig 1) and ASW-BTN vs untreated BTN (F-G, as seen in Fig 4).** All genes are shown as grey points, and in each panel black points mark those genes known to be associated with human inborn errors of immunity (**A**), copy number over 5 (**B**), copy number under 2 (**C**), Steamer insertions within 2kB upstream (**D**, **F**), or gene fusions (**E**, **G**).
(TIF)

**S5 Fig. Hierarchical clustering separates samples by seawater treatment.** Hierarchical clustering using all expressed genes for untreated healthy hemocytes ("heme" - blue), ASW-treated healthy hemocytes ("heme_ASW" - pink), untreated MarBTN ("BTN" - red) and ASW-treated MarBTN ("BTN-ASW" - green). Samples are labeled by their source clam and treatment.
(TIF)

**S6 Fig. Differential expressed pathways show similarities across comparisons.** Adjusted p-values, further adjusted to be positive for upregulation and negative for downregulation, to compare gene set enrichment analysis results from (**A**) ASW-treated MarBTN and untreated MarBTN each versus hemocytes, (**B**) ASW-treated versus untreated for each of MarBTN and hemocytes, and (**C**) the BTN response to seawater compared with the initial BTN/hemocytes results. In (**A**) results are highly correlated, with the same top up- and downregulated pathways regardless of which treatment is compared to healthy hemocytes. In (**B**), gene set groupings corresponding to conserved and MarBTN-specific seawater responses from Fig 4 are labeled. In (**C**), very few gene sets exist in the outer quadrants, indicating little overlap between the significant gene sets in the two comparisons. Lines represent false discovery-corrected $p < 0.05$ significance thresholds.
(TIF)

**S1 Table. MAKER and NCBI gene annotation statistics.**
(XLSX)

**S2 Table. Sample list.**
(XLSX)

**S3 Table. BTN vs Hemocytes upregulated genes.**
(XLSX)

**S4 Table. BTN vs Hemocytes downregulated genes.**
(XLSX)

**S5 Table. BTN vs Hemocytes upregulated gene sets.**
(XLSX)

**S6 Table. BTN vs Hemocytes downregulated gene sets.**
(XLSX)

**S7 Table. MarBTN and MtrBTN gene regulation comparison.**
(XLSX)

**S8 Table. Fusion genes.**
(XLSX)

**S9 Table. ASW-BTN vs BTN upregulated genes, conserved in hemocytes.**
(XLSX)

**S10 Table. ASW-BTN vs BTN upregulated gene sets, conserved in hemocytes.**
(XLSX)

**S11 Table. ASW-BTN vs BTN downregulated genes, conserved in hemocytes.**
(XLSX)

**S12 Table. ASW-BTN vs BTN downregulated gene sets, conserved in hemocytes.**
(XLSX)

**S13 Table. ASW-BTN vs BTN upregulated genes, specific to MarBTN.**
(XLSX)

**S14 Table. ASW-BTN vs BTN upregulated gene sets, specific to MarBTN.**
(XLSX)

**S15 Table. ASW-BTN vs BTN downregulated genes, specific to MarBTN.**
(XLSX)

**S16 Table. ASW-BTN vs BTN downregulated gene sets, specific to MarBTN.**
(XLSX)

**S17 Table. Gene sets.**
(XLSX)

**S1 Data. Data A in S1 Data, Data for Fig 1a: List of differentially expressed genes in MarBTN. Data B in S1 Data, Data for Fig 1b: List of differentially expressed gene sets in MarBTN. Data C in S1 Data, Data for Fig 2a: Numbers of fusion genes in each sample. Data D in S1 Data, Data for Fig 2b: List of differentially expressed genes in MarBTN by copy number. Data E in S1 Data, Data for Fig 2c: Differential expression of genes with *Steamer* insertion in the gene or within 2kb upstream. Data F in S1 Data, Data for Fig 3c: Values of principle components 1 and two for the analysis of transcriptomic response**

to seawater. **Data G in** S1 Data**, Data for** Fig 4a**: Genes differentially expressed between MarBTN and ASW-treated MarBTN that are also differentially expressed between healthy hemocytes** and **ASW-treated healthy hemocytes. Data H in** S1 Data**, Data for** Fig 4b**: Gene sets differentially expressed between MarBTN** and **ASW-treated MarBTN that are also differentially expressed between healthy hemocytes** and **ASW-treated healthy hemocytes. Data I in** S1 Data**, Data for** Fig 4c**: Genes differentially expressed between MarBTN and ASW-treated MarBTN that are not differentially expressed between healthy hemocytes** and **ASW-treated healthy hemocytes. Data J in** S1 Data**, Data for** Fig 4d**: Gene sets differentially expressed between MarBTN and ASW-treated MarBTN that are not differentially expressed between healthy hemocytes and ASW-treated healthy hemocytes.**
(XLSX)

## Acknowledgements

We thank Metzger lab members Jordana Sevigny, Karyn Tindbaek, and Finola Schmahl-Waggoner for feedback, Marisa Yonemitsu for dissecting the original reference clam, and Sophie Kogut for investigating fusion genes.

## Author contributions

**Conceptualization:** Samuel F.M. Hart, Michael J Metzger.

**Data curation:** Samuel F.M. Hart.

**Formal analysis:** Samuel F.M. Hart.

**Funding acquisition:** Michael J Metzger.

**Investigation:** Samuel F.M. Hart, Fiona E.S. Garrett, Jesse S Kerr, Michael J Metzger.

**Methodology:** Samuel F.M. Hart, Michael J Metzger.

**Project administration:** Samuel F.M. Hart, Michael J Metzger.

**Resources:** Michael J Metzger.

**Supervision:** Michael J Metzger.

**Validation:** Samuel F.M. Hart, Michael J Metzger.

**Visualization:** Samuel F.M. Hart, Michael J Metzger.

**Writing – original draft:** Samuel F.M. Hart, Michael J Metzger.

**Writing – review & editing:** Samuel F.M. Hart, Fiona E.S. Garrett, Jesse S Kerr, Michael J Metzger.

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
