## [Decision Letter · Decision Letter 0]

21 Nov 2024

PGENETICS-D-24-01028Gene expression in soft-shell clam (Mya arenaria) transmissible cancer reveals survival mechanisms during host infection and seawater transferPLOS Genetics Dear Dr. Metzger, Thank you for submitting your manuscript to PLOS Genetics. After careful consideration, we feel that it has merit but does not fully meet PLOS Genetics's publication criteria as it currently stands. Therefore, we invite you to submit a revised version of the manuscript that addresses the points raised during the review process. Please submit your revised manuscript within 60 days Jan 20 2025 11:59PM. If you will need more time than this to complete your revisions, please reply to this message or contact the journal office at plosgenetics@plos.org. Please include the following items when submitting your revised manuscript:* A rebuttal letter that responds to each point raised by the editor and reviewer(s). You should upload this letter as a separate file labeled 'Response to Reviewers'. This file does not need to include responses to any formatting updates and technical items listed in the 'Journal Requirements' section below.* A marked-up copy of your manuscript that highlights changes made to the original version. You should upload this as a separate file labeled 'Revised Manuscript with Track Changes'.* An unmarked version of your revised paper without tracked changes. You should upload this as a separate file labeled 'Manuscript'. If you would like to make changes to your financial disclosure, competing interests statement, or data availability statement, please make these updates within the submission form at the time of resubmission. Guidelines for resubmitting your figure files are available below the reviewer comments at the end of this letter. We look forward to receiving your revised manuscript. Kind regards, Kent W. HunterSection EditorPLOS Genetics Kent HunterSection EditorPLOS Genetics

Aimée Dudley

Editor-in-Chief

PLOS Genetics

Anne Goriely

Editor-in-Chief

PLOS Genetics

  **Journal Requirements:**

1) Please provide an Author Summary. This should appear in your manuscript between the Abstract (if applicable) and the Introduction, and should be 150u2013200 words long. The aim should be to make your findings accessible to a wide audience that includes both scientists and non-scientists. Sample summaries can be found on our website under Submission Guidelines:

https://journals.plos.org/plosgenetics/s/submission-guidelines#loc-parts-of-a-submission

4) We notice that your supplementary Figures are included in the manuscript file. Please remove them and upload them with the file type 'Supporting Information'. Please ensure that each Supporting Information file has a legend listed in the manuscript after the references list.

 **Reviewers' comments:** Reviewer's Responses to Questions

**Comments to the Authors:**

Reviewer #1: In nature, there are three types of naturally transmissible cancers; one affects dogs (canine transmissible venereal tumour or CTVT), two affect Tasmanian devils and several affect soft-shell clams. The cell identity of these transmissible cancers is poorly understood, which limits our knowledge of the origin and evolution of these extraordinary tumours. In this paper, Hart et al. characterise the transcriptional signature of Mya arenaria bivalve transmissible neoplasia or BTN. The authors performed RNAseq to compare normal clam hemocytes and BTN samples and found a common core signature, supporting the notion that BTN emerged from hemocytes, but also found a substantial number of differentially expressed genes DEGs by gene set enrichment analysis (GSEA). Notably, they detected downregulation of innate immune pathways in BTN, suggesting a mechanism for immune evasion. The authors found increased copy number variation in BTN, which correlated with changes in expression. Notably, when the transcriptional profiling was repeated in the presence of sea water to mimic the extra-host life cycle of BTN or hemocytes, these same innate immune genes were found specifically upregulated in BTN relative to hemocytes, suggesting a potential function in transmission and/or survival.

The results are interesting and point to an important and potentially opposite, function of innate immune pathways in regulating BTN growth and transmission. Characterising the physiological response of hemocytes and BTN to seawater is also of interest. The results are well presented and for the most part convincing, however there are elements of the paper that could be improved.

1. The study is entirely based on RNAseq and some confirmation of key results by different methods would strengthen the conclusions. For example, qPCR on a selected array of genes, or, when suitable antibodies are available, Western blotting or flow cytometry to detect protein levels.

2. GSEA is a valuable, but it could be complemented by alternative methods such as Ingenuity Pathway Analysis, or other suitable bioinformatics approaches.

3. In Supplementary Fig. 4, the “ASW” should be labelled “BTN-ASW” for clarity. In the heatmap, differences between groups are not that clear. Could perhaps an additional clustering based on Log2FC or adjusted p be added?

4. In Supplementary Tables, the genes included in each GSEA pathway should be shown to enable better comparisons.

5. Did the authors test is sea water induced greater activation of Steamer, or other ERVs? Often, activation of ERVs is detected by nucleic acids sensors triggering innate immune activation.

6. Can the authors map the genomic copy number (Figure 2B) to the genes included in the inflammatory pathways detected by GSEA?

Reviewer #2: In this article, the authors quantify and analyse gene expression in a transmissible cancer impacting soft-shell clams. The authors are looking to identify genes and pathways that appear to be central for the parasitic ability of these BTNs, which will allow them to get more insights into the processes that drive a cell to become a transmissible cancer. They perform a well-executed differential expression analysis comparing the transmissible cancer to its supposed tissue of origin – healthy hemocytes. They go further by experimentally subjecting hemocytes and transmissible cancer cells to seawater treatment and observing the responses in terms of genes expressed. Their set up and comparison is useful to distinguish genes whose expression varies in both hemocytes and BTN, from genes that only seem to be up or down regulated specifically in BTN. The authors did a good job of summarizing their findings of genes up or down regulated, and do not overstate their conclusions. Some more precisions on technical aspects and statistics would be appreciated to clarify the methods, as well as a better explanation and annotation of the figures which would permit a quicker understanding of the results and discrimination of what is up and down regulated among an extensive number of genes presented.

Here are the details of what would need to be clarified in order to improve this paper:

In the text:

1- The annotation contains 44,373 gene models (according to supplementary table 1). Could the authors possibly state this number in the text, discuss why this is higher than would be expected, and explain that all gene models (44,373 genes) were used in their DEseq analysis.

2- P.3 l.13: Please write out what BUSCOs are as well as a quick explanation in the main text.

3- P.4 l.14: Here in particular (for TENX and CNTN5) and at a few other places, it is not immediately clear in the text if the genes being mentioned are up or down regulated. Specifying that would be helpful.

4- p.5 l.2 “which order ranks genes” – explain what this means/how this is done.

5- P.6 l.7-17: It is to be commended that the authors are careful with interpretations and raise the possibility that the observation of downregulation of immune genes does not automatically mean that those represent and evolutionary response for the cancer as a strategy to evade hosts response. However, the comparison with a pool of other tissues as explained in the paper does not seem to entirely suffice to be convinced that the differences observed between hemocytes and BTN cells solely reflect mechanisms selected for immune evasion/driving cancer. In particular, have the authors considered the possibility that this cancer could have emerged from hemocytes at a different differentiation stage than the fully mature hemocytes used for comparison, and that the differences in some pathways could reflect those differences in cell state/developmental stage?

6- P.6 l.22: Please provide a little more detail about the study in MtrBTN2 by Burioli et al. to which the genes and pathways are compared, in particular, was this also a BTN vs hemocytes comparison?

7- P.7 l.2: Specify the degrees of freedom for the chi-squared test.

8- P.9 l.4-9: The first part about Steamer insertion and the comparison when it is inserted within genes is clear, but what follows with the more insertions within 2kb is a bit confusing. It is not very clear how the comparison is done and what exactly is being compared to what (median fold change? What are the groups and the null hypothesis?). Because of that, it is not straightforward what the p-values in the text are representing (p=0.038 for which comparison and which null hypothesis? One of the labels in the figure suggests that ** p< 0.005, how does that compare?). Could the relatively low sample size in the control group (17) decrease the robustness of the result? Overall, it would be helpful if this part were clarified as it is for now not clear how it confidently contributes to ruling out the hypothesis that increased expression and Steamer could be linked to chromatin accessibility/pre-existing high expression instead of a true causation.

9- P.11 l.24: How many genes/ gene sets were up/down regulated in each case?

10- P.12 l.9 Given all the data that the authors have gathered, it could be nice to go a bit further and be more precise about the genes and pathways that have a “reversal of the downregulation observed in untreated MarBTN cells” – How many? Which ones are they specifically? There is probably a way of having a better visualization of for which of these exact pathways this is the case.

11- P. 13 l8-9. “These responses may reflect metabolic processes that are active in proliferative MarBTN, but not hemocytes, that are shut down in response to seawater exposure”: Could the authors comment on whether these specific genes and pathways are upregulated in MarBTN vs hemocytes in their initial (not seawater) comparisons? This would help to support their hypothesis.

12- P.16 l12: Processed data outputs, i.e. gene tables and pathway tables used to create figures, should be supplied as accompanying source data to improve reproducibility (please could this be provided for data underlying Figures 2B and 2C?).

13- P.17 l.11-13: How certain is it that the cells adhering to the plate are only host hemocytes? Could this protocol of collecting the non-adherent cells be biasing the cell population? Is there a reference for this?

14- P.17 l.15: Could antibiotics influence the gene expression response? Is there a way to control for that?

15- P.18 l.9: Please provide more details for the sequencing approach: read length, insert size, paired or single end?

16- P.18 l.16: It might be good to specify that DESeq2 includes a library size normalization in its model to know that it has been done at some step of the analysis.

17- P.18 l.22-23: Some clarifications are needed regarding the normalization, is it to the maximum or minimum sample? Was this done before DESeq2?

18- P.19 l.11-15: What proportion of genes could be determined/annotated in this way? Could some of these categories just have 1 gene in them for example? Can it exaggerate the statistical significance? It is not clear how robust the method would be for custom gene sets determined that way.

19- P.19 l.15: More information on how the ranking is established and what the parameters mean would be helpful.

20- Supp T1: not sure what the letters (S) (D) etc. mean, although I guess M for Missing etc.

21- Supp T4: the gene corresponding to the dot labelled CNTN5 in fig1A seems to be named NPHN, are they the same?

22- Supp T7: what is the agg_fold change?

In the figures:

23- Figure 1: According to their description in the text, 60% genes (corresponding to ~26k genes) were not significantly up or downregulated. These are not visible in the plot. Please could this be clarified.

24- Figure 1 B (+) regulation of… means gene involved in positively regulating given pathway? Specify in legend.

25- Figure 1 and 4 need better annotating on the plot itself to understand what part of the plot is up and downregulated. In figure 4 add some labels directly on the plots to understand that A-B= conserved set of genes / C-D = MarBTN specific.

26- For figure 2-B, it would be useful to plot the data as well as the boxes, the results seem very nice and convincing it would be great to see the actual data points that go with them. Also in the legend specify that we look at the median fold change (if it is the case).

27- Figure 3, add in legend that ASW= artificial sea water.

28- Supplementary figure 2: Define the red dotted line, and explain how the ranked list of genes is obtained.

29- Supplementary figure 3 and 5: although providing and appreciated control and being obviously well thought through, these figures still result a bit hard to read, some more annotation directly on the figure “quadrants” to make clear what is up and down regulated in what would be helpful.

30- Supplementary figure 4: What exactly is the gene set in this figure? It looks like a fewer number than the entire set of genes, as seems to be stated in the legend.

Additionally, a few typos/reformulations:

p.7 l.22 “Fusions found all cancer samples” – word “in” missing

p. 12 l 3-6: “The response to glucose/oxygen deprivation and negative regulation of apoptotic processes are among the top upregulated gene sets (Supplementary Table 14), indicating that these survival responses may be more broadly MarBTN-specific across additional genes.”: I am not sure I understand what the last part of the sentence means.

p. 19 l.13 “leaving excluding genes” - remove one verb.

Reviewer #3: This manuscript describes changes in gene expression in the Bivalve Transmissible Neoplasm from Mya arenaria, compared to its cell type of origin, a Hemocyte, and when it moves through an essential part of its life cycle, passage through seawater. As expected, the authors find vast number of genes significantly up and down regulated and highlight some of the most significant differences, which describe key aspects in the cancers ability to transmit. Interestingly, immune system pathways are down-regulated in MarBTN compared to the ancestral Hemocyte, indicating immune evasion. A subset of the gene expression changes are due to copy number variation (in an unstable genome) and the insertion of a retrotransposon, steamer. The most intriguing experiments for myself are the mimicry of sea water transmission and the plasticity in gene expression that allows a sedentary state for the cells. Overall, the paper is well written and clear and provides insights into cancer cell plasticity and transmission of interest across the cancer biology field. I have several queries and additions below:

• Page 8 – in the description of the genes up and down regulated it would be helpful for the reader if the log-fold change and p-value for each gene was provided in parentheses. It would give some context when read with Figure 1.

• Copy number variation is clearly playing a role in gene expression changes as the statistical tests and the number of DEGs show in 2B. What proportion of the DEGs does copy number variation explain?

• Similarly for the Steamer insertion, does steamer and CNV account for most of the expression change? This could be expressed in the results.

• For the major genes identified in Figure 1, are these the result of CNV or steamer? If not, have the mechanisms been investigated?

• The plasticity identified as the cells move into seawater is very interesting and could be explored further using the analysis on CNV and steamer above. For example, for the immune system genes de-regulated compared to Hemocytes and in sea water, are the mechanisms the same? There could be further explanation of this in the discussion.

• Line 13 page 13 – Join sentence to preceding paragraph

**Have all data underlying the figures and results presented in the manuscript been provided?**

Reviewer #1: **No: **GSEA gene sets not provided

Reviewer #2: **No: **Please provide source data underlying Figure 2B, 2C

Reviewer #3: Yes

PLOS authors have the option to publish the peer review history of their article (what does this mean?). If published, this will include your full peer review and any attached files.

Reviewer #1: **Yes: **Ariberto Fassati

Reviewer #2: No

Reviewer #3: **Yes: **Dr Hannah Siddle

---

## [Decision Letter · Decision Letter 1]

18 Feb 2025

Dear Dr Metzger,

We are pleased to inform you that your manuscript entitled "Gene expression in soft-shell clam (Mya arenaria) transmissible cancer reveals survival mechanisms during host infection and seawater transfer" has been editorially accepted for publication in PLOS Genetics. Congratulations!

Yours sincerely,

Hongbin Ji

Section Editor

PLOS Genetics

Kent Hunter

Section Editor

PLOS Genetics

Aimée Dudley

Editor-in-Chief

PLOS Genetics

Anne Goriely

Editor-in-Chief

PLOS Genetics

**Data Deposition**

http://datadryad.org/submit?journalID=pgenetics&manu=PGENETICS-D-24-01028R1

**Press Queries**

---

## [Editor Report · Acceptance letter]

PGENETICS-D-24-01028R1

Gene expression in soft-shell clam (Mya arenaria) transmissible cancer reveals survival mechanisms during host infection and seawater transfer

Dear Dr Metzger,

We are pleased to inform you that your manuscript entitled "Gene expression in soft-shell clam (Mya arenaria) transmissible cancer reveals survival mechanisms during host infection and seawater transfer" has been formally accepted for publication in PLOS Genetics! Your manuscript is now with our production department and you will be notified of the publication date in due course.

With kind regards,

Anita Estes

PLOS Genetics

On behalf of:
